

# Tectonic controls on the ecosystem of the Mara River Basin, East Africa, from geomorphological and spectral indices analysis

Alina Lucia Ludat[1], Simon Kübler[1]

[1] Department of Earth and Environmental Sciences, Ludwig Maximilian University of Munich, Munich, 80333, Germany

*Correspondence to*: Alina Ludat (alina.ludat@outlook.de), Simon Kübler (s.kuebler@lmu.de)

**Abstract.**

Tectonic activity impacts the environment, therefore, identifying the influence of active faulting on environmental factors, such as soil development and vegetation growth patterns, is valuable in better understanding ecosystem functions. Here, we illustrate how tectonic activity and lithology of bedrock influence temporal and spatial patterns of vegetation and soil

parameters in a fault-controlled river basin.

The Mara River Basin lies in a region of previously unrecognised active normal faulting, dominated by the Utimbara and Isuria faults, resulting in areas of relative uplift, subsidence and tilting. Faulting leads to spatially variable erosion and soil formation rates as well as disruption and modification of drainage systems. On a small scale, steep escarpments cast shade and provide shelter. All of these factors might be expected to exert controls on ecosystem dynamics on a range of lengths and

timescales. Here, we investigate tectonic controls on ecological processes in the Mara River Basin using TanDEM-X and Sentinel-2 data. We use high-resolution digital elevation models (DEMs) to map the Utimbara and Isuria faults and to measure the height of the escarpments (up to 400 m) along the length of the faults. Total fault offset can be estimated by correlating Neogene phonolite lavas (thought to be 3.5 - 4.5 Ma old) on either side of the faults. If the age is correct, slip rates can be estimated to be on the order of 0.1mm yr[-1]. Analysis of DEMs also reveals the presence of recent earthquake scarps in the

hanging wall sediments of the main faults and extensive alluvial fan formation on the hanging wall. Low mountain front sinuosity values and the presence of steep escarpments also suggest recent activity. Drainage is displaced across the fault traces, and, in one area, it is possible to map the lateral channel migration of the Mara River due to hanging wall tilting.

We used a 5-year Normalised Difference Vegetation Index (NDVI) time-series, Clay Mineral Ratio (CMR) and Moisture Stress Index (MSI) to investigate spatiotemporal vegetation patterns and soil formation. Whilst lithology does exert some

control, as expected, we observed that the downthrown hanging wall of the faults, especially directly adjacent to the escarpment, is consistently associated with a higher degree of vegetation, wetland formation and clay distribution. Analysis of spectral indices shows that the overall spatial pattern of vegetation cover is seasonally low in the flat plains and perennially high in the vicinity of more complex, tectonically influenced structures. The NDVI highlights several locations with permanently healthy vegetation along the escarpment which extend downslope for several kilometres. Our study shows that in

the Mara River Basin, active normal faulting is an important stabiliser of vegetation growth patterns, likely caused by favourable hydrological and pedological conditions along the escarpments; tectonic activity has a direct beneficial influence on ecological processes in this climatically sensitive region.

## 1 Introduction

Landscapes in tectonically active settings host some of the most complex ecosystems in the world (e.g. Shlemon and Riefner,

2006), so research in such settings bridges a wide array of fields including geology, hydrology, and ecology. Recent studies have shown that tectonic activity can have both, beneficiary and disadvantageous, effects on ecosystems (Kübler et al., 2021; Veblen et al., 2016; Reynolds et al., 2016). In extensional regimes, subsidence of the hanging wall provides accommodation space for sediments and water. Vertical surface motion relative to the water table leads to formation and/or modification of



drainage systems and landforms that account for a significant hydrological complexity, such as inland lakes and wetlands
(Bailey et al., 2000; Bishop, 1995; Forsberg et al., 2000). In addition, fracturing along fault zones can create permeable
channels that promote water circulation, resulting in hydrological surface features like springs and swamps in the vicinity of
active faults. Zones of seasonally stable surface and subsurface water control the formation of climatically insensitive
vegetation cover and create potential refugia for wildlife and livestock during dry periods (Sinclair et al., 2008b; Swallow et
al., 2009). Tectonic footwall uplift leads to increased erosion of the fault escarpment and uplifted footwall block, resulting in
localized downslope improvement of the nutritional quality of the soil due to the transport of unweathered bedrock material
(Porder et al., 2005; Carey et al., 2005). The combination of hydrological stability and increase in soil quality allows for
reliable conditions for vegetation growth and for agricultural land use in the hanging wall regions (Swallow et al., 2009).
Conversely, increased erosion rates along and above the fault scarps leads to soil degradation and may be associated with
poorer vegetation and unreliable agricultural productivity on the uplifted footwall (Pal et al., 2009).

The objective of this study is to assess the influence of tectonic activity on temporal and spatial vegetation patterns in a fault-
controlled river basin of the East African Rift System (EARS). We explore the relationship of tectonic landforms with NDVI
(Normalised difference vegetation index), MSI (Moisture stress index) and CMR (Clay mineral ratio) using a range of
geospatial and remote sensing approaches. The combination of three spectral indices as proxies for vegetation stability and
soil quality allows conclusions about ecosystem functions. A deeper understanding of the influence of tectonic activity on
hydrological, pedological and biological properties provides a new perspective on the variety of abiotic, particularly geological,
factors on stabilizing the distribution and availability of fertile soils and vegetation in sensitive ecosystems.

The EARS is a tectonically active landscape, characterized by a complex geological and geomorphological setting and a high
biodiversity (Ring et al., 2018). Landscape evolution of the EARS is controlled by continental rifting processes, including
uplift, volcanism and earthquake faulting as well as erosional and depositional processes (Burbank and Pinter, 1999; Bailey et
al., 2000). However, little research to date has focused on the effect of tectonic activity and subsequent landscape evolution
on ecosystem dynamics (Kübler et al., 2021; Bicudo et al., 2019; Veblen et al., 2016). Particularly, the role of tectonic surface
faulting on spatiotemporal vegetation patterns remains enigmatic. The limited knowledge of how tectonic processes influence
vegetation stability in regions with highly variable climate is a significant gap. Tectonic processes potentially contribute to
stabilizing conditions for rainfed agriculture and conserving nature in the face of environmental and climate change (Comer
et al., 2015).

The transboundary Mara River Basin (MRB) in Kenya and Tanzania is located between the eastern and western branches of
the EARS and as such represents a key region to study the connection between active tectonics, geology, ecosystems processes,
and human-landscape-interaction. The basin has received much attention from both researchers and tourists due to a rich
biodiversity associated with the Masai Mara Reserve and Serengeti National Park (Mcclain et al., 2014; Sinclair et al., 2008b)
and is home to the world's most thoroughly documented ungulate migration (Maddock, 1979; Pennycuick, 1975). While the
ecoclimatic aspects of the region, such as rainfall patterns (Bartzke et al., 2018; Norton-Griffiths et al., 1975), soil processes
(Jager, 1982), vegetation dynamics (Reed et al., 2009) and large mammal migrations (Belsky, 1986) have been intensively
studied, there is a lack of studies focussing on the impact of geotectonic processes on soil, vegetation and fauna dynamics. In
a geological context, the MRB has mainly been explored for mining purposes (Henckel et al., 2016; Smith and Anderson,
2003). The Quaternary-to-recent tectonic activity of the region, let alone the interactions between active faulting, bedrock
lithology, fluvial processes and vegetation growth have not been examined in detail. Thus, tectonic structures and the timing
of tectonic activity in the basin are not well documented on the twelve available geological maps (Grey and Macdonald, 1966;
Grey et al., 1969; Jennings, 1966; Mulgrew, 1966; Saggerson, 1966; Thomas, 1968; Thomas and Kennedy, 1977; Williams,
1964b, 1969, 1964a; Wright, 1966).



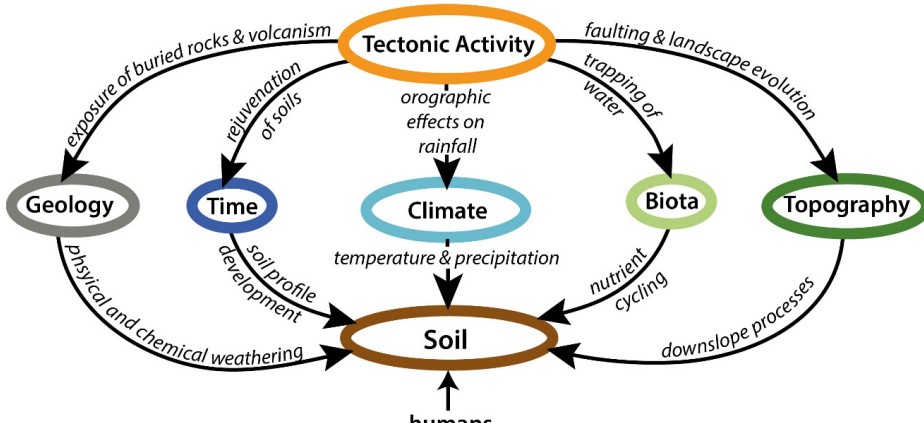

**Figure 1 Tectonic activity has a direct influence on the major soil-forming factors, namely: parent material (geology), time, climate, organisms (biota) and topography defined by Jenny (1994).**

Soil formation is controlled by long-term ($10^3$-$10^6$ yr) factors such as, tectonic activity, geology, topography and climate, and factors acting on shorter time scales ($10^1$-$10^3$ yr) such as hydrological, biological and anthropogenic processes (Fig. 1). Active faulting leads to the exposure of buried bedrock, which then can be physically and chemically weathered to mineral-rich soils (Vitousek et al., 2003). Tectonic disruptions of the landscape also lead to rejuvenation of soils and over time influence the soil profile development and subsequently vegetation patterns. High escarpments from tectonic activity create orographic effects directly influencing precipitation patterns, which are a crucial part of soil-formation (Rohrmann et al., 2016). Micro-climatic changes play an important role during soil formation especially in regions with high amounts of rainfall and high temperature gradients like the MRB (Glover and Williams, 1966). Hence, it is important to analyse the tectonic and geological processes in the dynamic landscape of the MRB as they influence soil fertility, which in turn influences vegetation distribution, animal migration and agriculture.

## 2 Setting of the study area

### 2.1 Tectonic and geological setting

The MRB is located at the northeastern margin of the Tanzania Craton on the plateau between the eastern and western branch of the EARS, east of Lake Victoria (Fig. 2a). It gently slopes from the Mau escarpment (2940 m a.s.l.) toward Lake Victoria (1130 m a.s.l.) (Sinclair et al., 2008b) and extends from longitude 33° 47′ E to 35° 47′ E and from latitude 0° 38′ S to 1° 52′S covering an area of approximately 14,000 km². The two perennial tributaries of the Mara River, Amala and Nyangores, drain the forested headwaters and join to form the perennial Mara mainstem at ca. 1695m a.s.l. (McClain et al., 2014). From there it flows through the Masai Mara National Park in Kenya and the Serengeti National Park in Tanzania into Lake Victoria at Musoma Bay (Fig. 2a).

The MRB is characterized by a unique heterogeneous geology being positioned on the northeastern margin of the Tanzania Craton at the intersection of Precambrian orogenic belts (Fig. 2b). The oldest rocks in the basin are the Archean granitic gneisses and amphibolites of the Tanzania Craton which are exposed in the center of the basin (Grey et al., 1969). The Precambrian metasediments and metavolcanics of the Nyanzian-Kavirondian Orogenic Belt (aka Lake Victoria goldfields) dominate the geology in the west of the basin (Henckel et al., 2016). Neogene to recent lavas and tuffs, including basalt, trachyte, and phonolite, unconformably rest on the Precambrian rocks in the north and center of the basin (Shackleton, 1946). Parts of the remaining flat-lying trachytic phonolite lava sheets extend above and below the Isuria and Utimbara escarpments. Quaternary sediment and soils dominate the southwest and northeast of the basin.





The Tanzania Craton has been tectonically stable from the Cambrian (530 Ma) until the onset of Neogene extensional tectonics, associated with the formation of the EARS (Barth, 1990). Rifting and subsequent rift shoulder uplift, associated with the eastern branch of the EARS, combined with hydrological and erosion-deposition processes, have given rise to a diverse array of landforms across the region. On a smaller scale, the assymmetric central and southern basins of the MRB are controlled by the Plio-Pleistocene escarpments of the NNE-SSW trending Isuria fault (aka (Soit) Olooolo and Siria; from now on: Isuria)

and the roughly E-W trending Utimbara fault (aka Utimbaru; from now on: Utimbara) (Shackleton, 1946). The Isuria escarpment extends from Gibaso in Tanzania in the SW over a length of ~ 90 km to Chebunyo in Kenya in the NE (Fig. 2a). The escarpment of the southeast-dipping fault varies in height from 100 m in the northeast to ca. 400 m in the southwest. The Utimbara fault has a surface expression measuring over ~ 100 km from the mouth of the Mara River into Lake Victoria at Musoma Bay in the west to the Tagari Hills in the east. The fault downthrows to the south with the uplifted blocks showing

slight tilting to the north and its escarpment height varies from 150 m in the east up to 400 m in the center. The largest displacement of ca. 400 m along both faults is constrained by a discontinuous cap of a Neogene phonolite lava at the top of the escarpment, identical to the lava outcropping at the base (Saggerson, 1966). In addition to the main escarpments, the area is cut by several smaller E-W trending faults (Henckel et al., 2016).

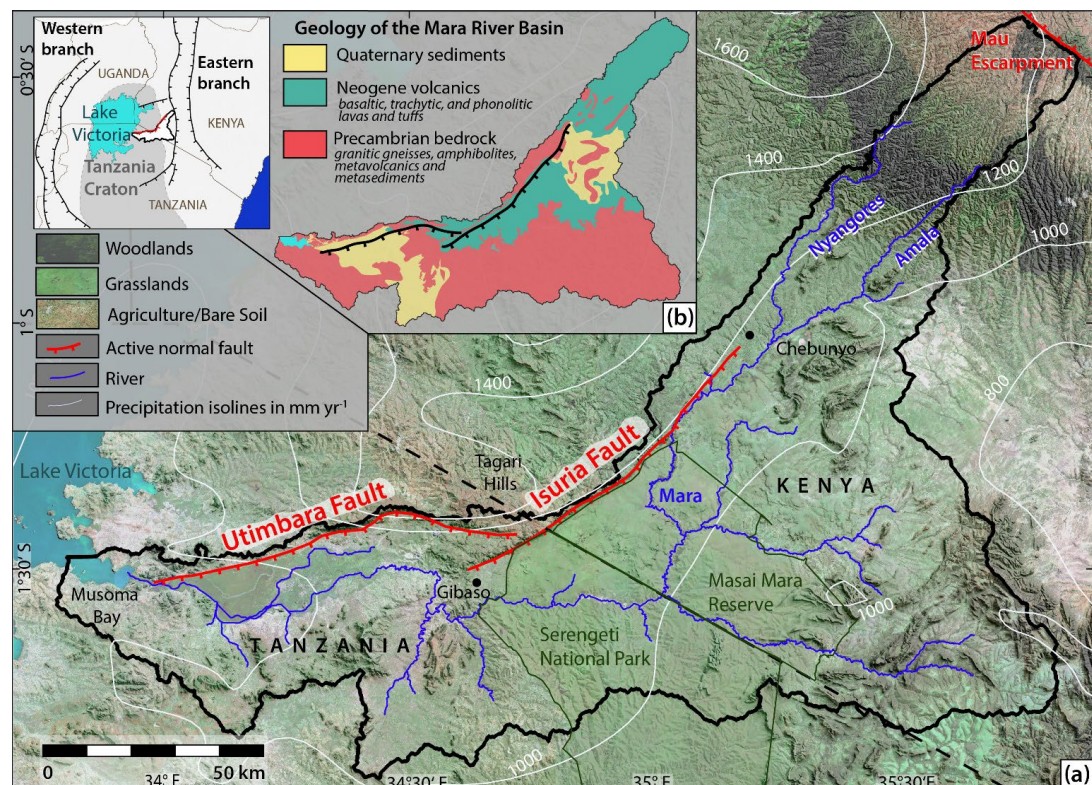

**Figure 2 Overview maps of the Mara River Basin in Kenya and Tanzania; (a) Vegetation types, major faults, rivers, precipitation rates, national park boundaries, and important place names of the study region. See small inset map of Eastern Africa for location. Vegetation derived from Sentinel-2 multispectral imagery from European Space Agency (ESA). Precipitation isolines derived from WorldClim version 2.1 by Fick and Hijmans (2017); (b) Simplified geological map derived from data by Geological Survey of Tanzania (GST); For simplification purposes all Precambrian rocks, all Neogene volcanics and Quaternary sediments were each**

**summarized as one unit. Thin, on the scale of the map not mappable, Quaternary deposits can be found all over the basin. Map projection for this and the following maps: UTM WGS 84 37 S.**

## 2.2 Climatic setting and vegetation

The MRB is located in a region of highly complex seasonal rainfall patterns and exhibits a bimodal rainfall with two rainy seasons linked to annual oscillations of the inter-tropical convergence zone in March-May (long rains) and October-December





(short rains) (McClain et al., 2014; Mwanake et al., 2019). Mean annual values of precipitation vary from 600 mm yr⁻¹ on the lowlands to nearly 1500 mm yr⁻¹ on the highlands (Fig. 2a) (Defersha et al., 2012). Significant differences in rainfall also occur between catchments on the escarpment and below the escarpment due to topographic effects and the regional influence of Lake Victoria (Camberlin et al., 2009). The mean daily temperature also changes with elevation, from 11 °C on the Mau Escarpment to 24° C down by Lake Victoria (Fick and Hijmans, 2017).

Fine-scale variation in soil properties in the MRB give rise to "catenas", soil sequences that vary from ridge top to hill slope, to foot slope, to valley. On ridge tops, soils are shallow and sandy, often exposing bedrock. Soils become progressively deeper and heavier further downslope, reaching maximum depths and levels of organic matter on valley floors (Jager, 1982). The granitic rocks and quartzites in the area are associated with grey to brown and sometimes reddish sandy soils (Cambisols). Residual soils on metavolcanics and metasediments are finer-grained, redder, and often lateritic (Mulgrew, 1966). Ferruginous

laterite and associated loams are generally found above and adjacent to the banded ironstone but can also develop on the Neogene volcanics. Black clay soils (mbuga) mainly develop on basaltic lavas, amphibolite and chlorite. Grey clastic alluvial deposits are mainly found in the Mara valley and surrounding riverbeds (Thomas and Kennedy, 1977). The soils from volcanic, fluvial, and colluvial origin are generally higher in Ca, Mg and plant-available P than those covering the rest of the system (Olff and Hopcraft, 2008).

The basin vegetation has undergone severe changes during the last century: the region was previously covered by montane forest in its headwater regions and a mixture of grasslands and shrublands in its middle and lower section. This pristine vegetation is today limited to the nature conservation areas of the Serengeti-Mara ecosystem (e.g., the Masai Mara National Reserve in Kenya and the Serengeti National Park in Tanzania) (Sinclair et al., 2008b). The main land cover in the basin can today be classified as agriculture (cropland and range lands), herbaceous vegetation and shrubland (savannah), residual

montane forest and wetland (Fao, 1994). Cultivated land accounted for approximately 35% of the basin in 2000, composed mainly of small-scale farms and tea plantations. Due to a continuing agricultural expansion and urbanization, today large-scale farms and irrigated crops are increasing in the area (Mcclain et al., 2014). Along the Mara River and its tributaries, gallery forest with closed canopy prevails (Sinclair et al., 2008b).

**3 Methodology**

We have applied a variety of remote sensing methods to analyse optical, topographic, and multispectral satellite data. The aim was to map tectonogeomorphological features and vegetation patterns along the Isuria-Utimbara Fault Zone (IUFZ). Remote sensing and geospatial analysis are potent tools for this purpose, as detailed analysis of the topography provides detailed information about the surface deformation history of inaccessible and remote areas (Pérez-Peña et al., 2010).

**3.1 Tectonic Geomorphology**

**3.1.1 Tectonogeomorphological Mapping**

We precisely mapped geomorphological features indicating tectonic activity based on the TanDEM-X digital elevation model (DEM) with 12 m horizontal resolution and 2 m relative height accuracy (Zink et al., 2017). From the DEM, we created hillshades for 3D-terrain and texture-shaded maps to emphasize structural features and the drainage network. Texture shading is a special technique for generating shaded relief images that show terrain features independently from their orientation, because it is not based on an illumination model. A strong light-to-dark transition at any cliff helps identifying fault scarps and

other linear features (Brown, 2014).

As the available geological maps of the study area do not allow a precise distinction between Precambrian faults and shears, minor fractures in granitic terrain, and young rift faults (Grey and Macdonald, 1966; Grey et al., 1969; Jennings, 1966; Mulgrew, 1966; Saggerson, 1966; Schoeman, 1947; Thomas, 1968; Thomas and Kennedy, 1977; Williams, 1964b, 1969,



1964a; Wright, 1966), we documented present tectonic activity by detailed fault-segment mapping. The IUFZ could be identified because the faults stand out in the landscape as steep escarpments accompanied by hydrological features such as hot springs and swamps on the downthrown side. In order to identify smaller fault scarps, we used offset geological contacts, downstream changes of drainage channels and changes in vegetation on ESRI World imagery (Esri et al., 2017). Furthermore, we explored the use of geomorphic markers, such as surface ruptures, offset drainages and fault-bounded alluvial fans which

could be related to recent fault activity and dated radiometrically in future field work.

### 3.1.2 Mountain-front sinuosity

Mountain front sinuosity is a well-established index in the field of tectonic geomorphology (Anand and Pradhan, 2019; Pérez-Peña et al., 2010; Soria-Jáuregui et al., 2019). It was determined to analyse the role of active tectonics in the deformational process and the topography of the basin as it allows an evaluation of tectonic activity along the mountain fronts. Its dependence

on climate and lithology (Bull, 1976) could be neglected as both factors do not vary much along the escarpments. Bull (1976) defined it as:

$$Smf = Lmf/Ls \tag{1}$$

where $Lmf$ is the length of the mountain front along the foot of the mountain and $Ls$ is the length of the mountain front measured along a straight line. Tectonically active mountain fronts usually produce straight fronts with low $Smf$ values as

uplift prevails over erosional processes. At less active or inactive fronts erosional processes generate sinuous fronts with high $Smf$ values. The $Smf$ of the investigated faults has been compared with values of other rift faults in the region. We used a $Smf$ value of lower than 1.4 to be indicative for a volcanics-capped tectonically active escarpment in the research area.

### 3.2 Morphometric analysis of the drainage network

We used the Copernicus DEM GLO-30 with 30 m horizontal resolution as input for the regional morphometric analysis of the

drainage basin. For this application, this DEM is superior to the higher-resolution TanDEM-X DEM because it is free of water-generated noise. Using QGIS (QGIS.org, 2021) with the SAGA plug-in (Conrad et al., 2015), the DEM was pre-processed to fill the sinks, and a flow accumulation raster was developed. The drainage network was calculated with a pixel threshold of 500 cells. To ensure the generated stream network is representative of the natural stream, drainages have been visually confirmed using high-resolution ESRI World imagery (ESRI et al., 2017).

### 3.3 Multispectral Analysis

Sentinel-2 multispectral imagery (Drusch et al., 2012) was used to extract landcover information. The Earth observation mission developed by the European Space Agency (ESA) acquires 13 bands in the VNIR (visible and near-infrared) to SWIR (short wavelength infrared) range, with four bands at 10 m, six bands at 20 m, and three atmospheric correction bands at 60 m spatial resolution (Drusch et al., 2012).

For the spectral indices, 69 Sentinel-2A scenes (granules T36MXD, T36MYD) with a cloud coverage lower than 5% acquired between 14th October 2016 and 11th January 2022 were selected. As Level-2A processed products only exist since December 2018, the previous data needed atmospheric, terrain and cirrus correction of Top-Of-Atmosphere using the Sen2Cor processor (Louis et al., 2016). A common method to process remote sensing data is producing band ratios. By taking the ratio of two bands or several bands the method highlights features of interest in the original grey scale image (Rowan et al., 1976; Inzana

et al., 2003). We used a combination of three spectral indices - Normalized Difference Vegetation Index (NDVI), Clay Mineral Ratio and Moisture Stress Index (MSI) – as a proxy for vegetation stability and soil quality, which allows conclusions about ecosystem functions. The NDVI was calculated from the 10 m resolution Sentinel-2 imagery to assess the spatial and temporal changes in primary production in the study area. NDVI is a measure of the amount of vegetation at the surface and is related to the health of the vegetation, as healthy (photosynthetically active) vegetation reflects a high amount of energy as compared



to the unhealthy and sparse vegetation (Grebby et al., 2014). In savannah environment plants include woody canopy, foliage and grasses (Sankaran et al., 2005).

The NDVI is calculated from the amount of red and near-infrared light reflected from the Earth's surface and gives values between 1 and -1. Higher values of NDVI indicate rich and healthier vegetation while the lower values indicate poor and sparse vegetation (Pettorelli et al., 2005). For Sentinel multispectral data, NDVI is calculated by: (B08 - B04) / (B08 + B04). The

average NDVI was calculated for the selected timespan. NDVI was used in this study as it is a reliable and well-studied ecological indicator. Other vegetation indices (GDVI, SAVI) were calculated but did not show significant differences compared to NDVI. In order to examine how NDVI varies along and perpendicular to strike, three longitudinal and 17 transverse profiles of both faults were extracted.

Additionally, a moisture stress index (MSI) was calculated from the 20 m resolution Sentinel imagery to determine vegetation

water stress and draw conclusions about plant available water. It was calculated as the ratio between the NIR and SWIR1 values (B11/B8). The values of this index range from 0 to over 3, whereas green vegetation commonly shows values between 0.4 to 2 (Hunt and Rock, 1989).

The clay mineral ratio (CMR) is a geological index, which helps identifying geological features containing clay and alunite as hydrous minerals absorb radiation in the shortwave infrared region (2.17 - 2.63 μm). It is calculated by the ratio between the

20 m resolution SWIR1 (B11) and SWIR2 (B12) bands (Alasta, 2011; Nath et al., 2019). As active fault zones are potential regions for primary clay mineral formation within the fault zone as well as accumulation of clay minerals through mineral weathering and down slope transport (Barton et al., 1995; Solum et al., 2005), this index can help identifying fault activity. However, this band ratio can also indicate carbonate mineralization and recently burnt areas, which demonstrate a high reflectance in the SWIR2 (Alasta, 2011).

**4 Results**

**4.1 Fault structures and tectonic geomorphology**

The morphology of the IUFZ is characteristic of a region of active normal faulting (Fig. 3), with an uplifted plateau on the footwall separated by a steep escarpment from a wide plain on the downthrown hanging wall. The plain is crossed by several smaller fault scarps mostly subparallel to the main escarpment. The EW-trending escarpment of the Utimbara fault can be

clearly traced on the satellite imagery over a length of ca. 65 km (Fig. 3a). A strongly eroded escarpment in the west towards Lake Victoria builds the western tip of the Utimbara fault adding to its total length of ca. 86 km. Where the Utimbara fault forms a stepover to the Isuria fault, a complex array of minor normal faults connects the two escarpments. The NE-SW trending escarpment of the Isuria fault is clearly visible on the TanDEM-X dataset over a length of ca. 86 km. If the eroded northern offshoot is included, the fault has a total length of ca. 93 km.






**Figure 3 Evidence for fault activity along the IUFZ with topography derived from TanDEM-X 12 m global digital elevation model © DLR 2018 (a) Mountain front sinuosity and slope map of the IUFZ including legend for all subfigures (b) Recent surface scarp along Utimbara escarpment with alluvial fans on texture-shaded digital elevation model (c) 3D topography model showing surface scarp (d) Recent surface scarp along Isuria escarpment with alluvial fans on texture-shaded digital elevation model (e) offset drainages along surface scarp on Google Earth imagery © Google Earth 2022.**

The striking Neogene escarpments of the Isuria and Utimbara faults are associated with surface displacements between 100 and 400 m measured from the TanDEM-X digital elevation model. The amount of vertical displacement varies over the length of the Utimbara escarpment. For our analysis, we focused on measurements along clear geomorphic expressions of the escarpment. From West to East, the height of the escarpment builds from ~ 100 m, increasing rapidly to 300 to 400 m in the centre. After a decrease down to ca. 100 m, the height increases again up to ca 230 m. The eastern offshoot of the fault slowly decreases in height from West to East. In general, the displacement curve shows a typical half-moon shape with the highest displacement in the fault centre.

From the southern tip towards the centre, the height of the Isuria escarpment increases constantly from ca. 50 m up to a maximum of ca. 390 m; towards the north the height decreases steadily and the escarpment ceases. The northern offshoots of



the Isuria fault show the smallest amount of vertical displacement. The total displacement could likely be higher than the measured values because, especially along the Utimbara fault, displacement markers are rare due to erosion of the Neogene volcanics, and the thickness of the volcanic rocks is poorly known.

The Isuria and Utimbara fault both show a continuous linear front (Figure 3 a), with minor interruptions because of the presence of normal faults oblique to the ridge. We split the escarpments into nine segments, here named with a first letter representing the main tectonic structure they belong to. Six different segments can be identified along the Utimbara escarpment: a strongly eroded discontinuous western front (U1), a less eroded front striking E-W (U2), a straight and continuous front striking E-W (U3), a slightly curved front striking E-W (U4), a continuous ESE-WNW striking front (U5) and, after a step-over, a slightly more eroded eastern front (U6). Three different segments can be identified along the Isuria front: a continuous southern front with a NE-SW main strike (I1), a continuous central front with a NNE-SSW strike (I2) and a less continuous northern front that shows N-S trending sections (I3). Smf index values were calculated for these nine mountain fronts. The highest value for the Utimbara front is 1.89, the lowest value is 1.25. Values for front sinuosity for the Isuria fault range from 1.06 to 1.2.

Long term fault displacement rates are bracketed by the exposed trachytic Phonolite lavas that are vertically displaced along the IUFZ by 390 (+/- 10) m. As the original geologic reports of the area predate the common use of radiometric age dating, Shackleton (1946) estimated these lavas to be of Pliocene (3.5 to 4.5 Ma) age, which still needs to be verified by radiometric dating. Field observations by Shackleton (1946) suggest an interval of tectonic quiescence after the eruption of the Neogene lavas, since quartz gravels occur on the lava surface, close to the edge of the Utimbara escarpment, south-west of Tarime. It therefore seems that the eruption of the Isuria lavas occurred some time before the onset of faulting activity, but the exact timing is yet to be determined. Assuming a maximum age of the onset of faulting of 3.5 to 3 Ma, the long-term displacement rate for both faults is approximately 0.10 mm y$^{-1}$, which falls within the typical range of fault slip rates in continental rift systems (e.g. Zielke and Strecker (2009) for EARS, Gold et al. (2017) for Lower Rhine Graben or Friedrich et al. (2003) for Basin and Range). The morphology of the Isuria escarpment, with Smf index values lower than 1.4 (Fig. 3a), indicates that also today tectonic activity exceeds erosion, especially in its central part (Anand and Pradhan, 2019; Pérez-Peña et al., 2010; Soria-Jáuregui et al., 2019). The Utimbara escarpment is similarly characterized by low Smf index values from 1.25 to 1.38 apart from two segments in the western portion showing values higher than 1.4. These segments coincide with the Mara Wetland, where alluvial deposits with high sedimentation rates occur.

Despite the subtropical climate and inferred low fault displacement rates of 0.1 mm y$^{-1}$, geomorphic indicators of the fault activity are preserved, likely because of the low erodibility of the Neogene volcanics. Tectonogeomorphological mapping along the IUFZ revealed several signs of recent surface rupturing activity. The most compelling geomorphological evidence for active faulting is the clearly traceable fault scarp ranging in height from approx. 2 to 35 m, which can be found in Pleisto-Holocene footwall sediments for several kilometres parallel to both escarpments of the IUFZ (Fig. 3b, c). The fault scarp segments are parallel to the escarpments and are frequently crossed by drainages, which show offsets when crossing the scarp (Fig. 3e). The offset horizontal offset is mostly perpendicular and varies from 10 to 20 m. The Isuria fault exposes five continuous fault scarps located within the colluvium on the hanging wall of the escarpment. Along the Utimbara fault we could determine a total of ten scarps. Slope profiles extracted from the DEM revealed a marked increase in slope angle from the tips of the Utimbara fault towards the centre.

We identified a series of small, fault-bound alluvial fans from the DEM exclusively on the Tanzanian side of the IUFZ (Fig. 3b, d). Nine inconspicuous alluvial fans could be identified along the Utimbara escarpment, directly underneath the escarpment. Along the Isuria escarpment 16 alluvial fans were found along the escarpment. The fans range in area from less than 0.1 km² to almost 0.5 km² and in thickness from 1 to 9 m as estimated from the DEM. They have a conical shape and are located close to the escarpment. The deposits along the Utimbara fault are generally thicker and larger than the deposits along the Isuria fault. At all fans sediment thickness appears to be highest in proximity to the fault scarp as is expected for fault-controlled alluvial fan deposits (Jackson and White, 1989). Where the rate of uplift is greater than the rate of drainage channel



erosion, providing a continuous supply of fresh debris, alluvial fans are deposited adjacent to the mountain front and continued tectonic uplift results in the accumulation of thick alluvial fan deposits (Bull, 1977). The detected series of sub-circular and not interlocked alluvial fans south of the IUFZ (Fig. 3b, e) provide geomorphic evidence for active extensional faulting predating alluvial fan deposition along the IUFZ. Given their limited extent and subtle geomorphic expression, tectonic activity likely proceeded in the Late Pleistocene or Early Holocene. Deposition ages of the deposits would enable a precise calculation of recent displacement rates.

Another indicator of ongoing tectonic activity are a number of wetlands and swamps throughout the basin, mainly concentrated in the river's floodplain (e.g. Mara Wetland). They form because of subsidence related to movement on the IUFZ and subsequent channel migration. Since the Mid-Pleistocene, tectonic activity significantly disrupted and reorganized the drainage network in the area leading to a periodic rejuvenation (Peters et al., 2008; Sinclair et al., 2008a). Approximately 2 km south of the North Mara goldmine, where the Mara River enters the Mara Wetland, the river channel visibly migrated towards the

Utimbara escarpment in the north (Fig. 7). It shows an asymmetric mosaic of meander loops and oxbow lakes indicating down tilt combing (Holbrook and Schumm, 1999). The further migration of the channel in tilt-direction though is inhibited by the massive granitic gneiss around the North Mara gold mine.

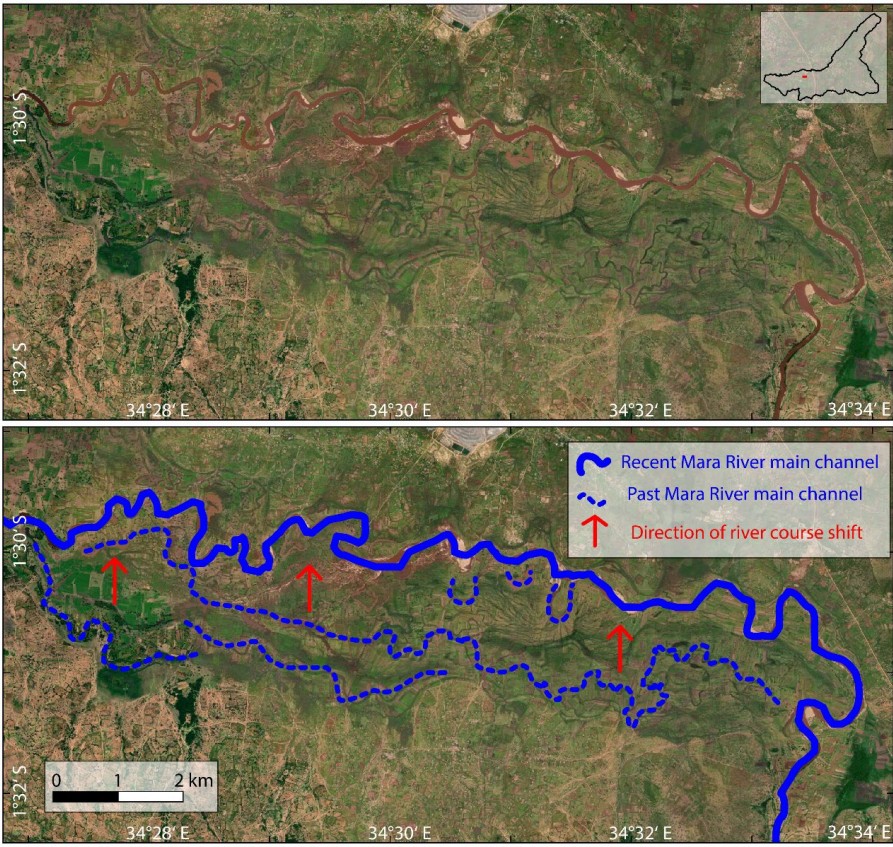

**Figure 4 Lateral northward channel migration of the Mara River due to tilt related with subsidence along the Utimbara fault, red**
**arrows indicate direction of tilt on Google Earth imagery © Google Earth 2022.**

Thermal springs often occur along tectonically active faults and are closely associated with the intensity and scale of modern activities of fault zones (Li et al., 2021; Delvaux et al., 2017). The association of the Isuria fault with the Majimoto hot spring (Fig. 3a) on the hanging wall therefore provides further evidence of current activity along the fault.



### 4.2 Spectral Indices

The NDVI in the study area varies from -0.22 to 0.84 with maximum values in the rainy season (March/April) (Fig. 5). Spatially, the NDVI also varies with low values in (barren) agricultural fields and grasslands and high values in forested areas and wetlands (Fig. 6a). The largest differences between rainy and dry season can be observed within the boundaries of the Serengeti National Park with dry season values varying between 0.2 and 0.4 and rainy season values from 0.6 to 0.8.

The average clay mineral ratio (CMR) in the study area varies from below 1 in water-covered areas to maximum values of

over 2.5 along the escarpments, in wetlands and forested areas (Fig. 6b, S1). Recently burnt areas show anomalously low values in dry season scenes (displayed in red). The CMR also reveals increased values along mapped fault scarps, as well as in areas dominated by volcanic rocks.

The average moisture stress index (MSI) values in the study area vary from 0.13 in the wetlands up to maximum values of 5.69 in agricultural fields (Fig. 6c, S2). Most pixels cluster around 0.5 to 1.5. Higher values indicate greater water stress and

less water content (displayed in red). The average moisture stress index revealed the highest moisture stress in the centre of the northern extension of the Serengeti National Park around the middle reaches of the Mara River. Similarly high values are found on agricultural fields on the Utimbara footwall.

Despite the complex precipitation patterns influencing vegetation distribution, the geological-tectonic signal can be extracted from spectral indices by time-series analysis. We evaluated a spectral signal time-series to separate the seasonal meteorological

from the long-term geological signal (Fig. 5, S1, S2). All indices highlight several locations with permanently favourable values for plant growth along the escarpments (Fig. 6a).

These include tectonic structures like the fault escarpments, the Mara Wetland south of the Utimbara fault and the marsh around the Majimoto Hot Spring south of the Isuria fault. Several smaller tectonic wetlands can be found along the Isuria fault exposing higher NDVI-values than the surrounding area. On a smaller scale, tectonically controlled alluvial fans on the hanging

wall appear on each of the indices (Fig. 6d, e, f) showing favourable conditions for plants to grow. Also, some tectonically quiescent, forested regions show favourable conditions for plant growth, such as the Ntagare Hills, the Mara gallery forest and the Trans Mara Conservation Area.



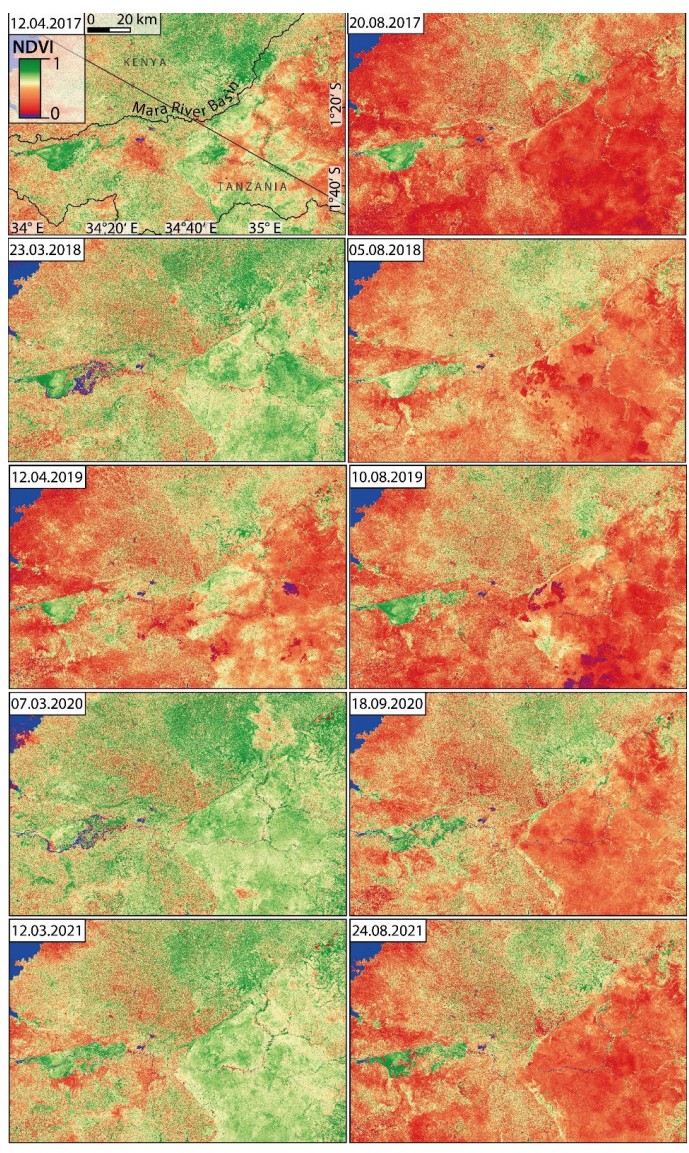

**Figure 5 Spatio-temporal distribution of NDVI in the Mara River Basin represented by 10 Sentinel-2 multispectral images from**
**European Space Agency (ESA). For each year a rainy season (left side) and dry season (right side) scene was selected to highlight areas with perennially stable vegetation (green).**



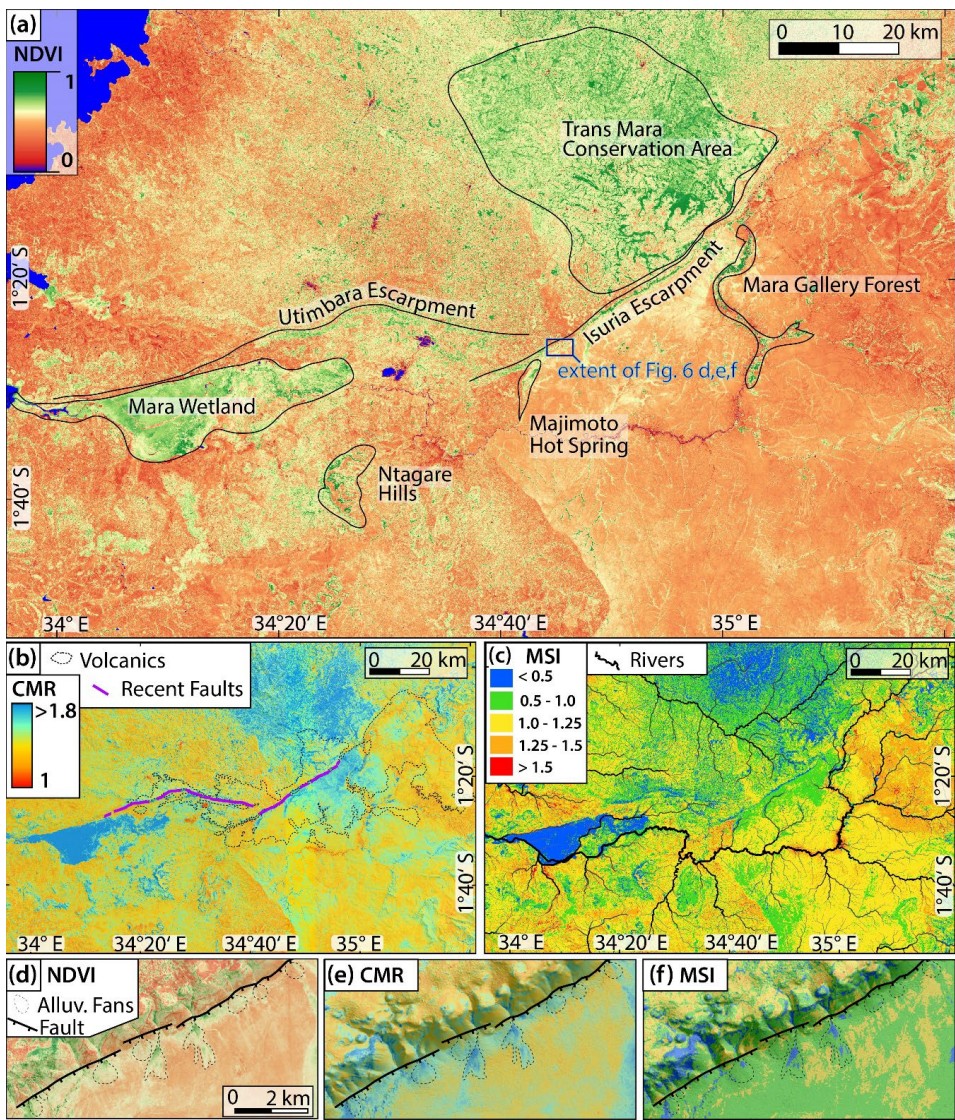

**Figure 6 Spectral indices along Isuria and Utimbara faults calculated from mean of 34 Sentinel-2 multispectral images from European Space Agency (ESA) acquired between 2016 and 2022. (a) Mean normalised difference vegetation index (NDVI); Locations with perennial stable vegetation are labeled and outlined. (b) Mean clay mineral ratio (CMR); volcanic rocks and recent fault scarps are indicated (c) Mean moisture stress index (MSI); river network is indicated. (d) Close-up of NDVI exposing higher values in fault-bound alluvial fans mapped from DEM (e) CMR showing clay accumulation in alluvial fans (f) MSI showing lower water stress in alluvial fans.**

## 5. Discussion

### 5.2 Geologic and tectonic control on ecosystem functions

#### 5.2.1 The effect of faulting on the hydrological network

Tectonic activity produces subsurface fractures and joints providing permeable pathways for fluids at various scales. Faulting plays an important role in controlling hydrology and developing drainage patterns (Holbrook and Schumm, 1999). which is in general also clearly observed along the IUFZ setting. The effect of faulting on hydrology can be seen clearly in the distribution



of wetlands on the hanging wall of the faults, hot springs along the fault traces and in the relatively high NDVI directly below the escarpments.

Perennial rivers and ephemeral streams often follow the geomorphic expression of fault zones, and in low-relief landscapes like the Serengeti plain, wetlands occur where the water table intersects the surface. The availability of surface water is dependent on climatic fluctuations, where seasonal precipitation absence can result in dramatic groundwater level changes up

to complete desiccation in extreme cases (Bailey et al., 2011). Generally, the large-scale flat land surface disruption by IUFZ actively leads to the formation and preservation of terraces and gorges on the footwall, as well as water and sediments accumulating on the hanging wall, facilitating the development of wetlands. The previously described northward migration of the Mara River channels due to block tilting is another important hydrological factor, as the abandoned meanders create important semi-stable water sources.

It is expected that if tectonic activity persists, the fault zone can maintain an almost permanent near-surface water table, and thus stable hydrological conditions and a climatically insensitive vegetation cover. Such a setting offers potential refugia for wildlife and livestock during dry periods. In the case of tectonic activity cessation the subsided regions will fill with sediment and the water table will drop (Bailey et al., 2011).

The marsh around the Majimoto hot spring (Fig. 6a) south of the Isuria fault is not only fed by perennial streams draining

towards the escarpment but is presumably related to hydrothermal activity along the Isuria fault. The area around the hot spring is clearly visible on the CMR (Fig. 6b) indicating it is a location of clay formation and/or accumulation. Maseke and Vegi (2019) measured temperatures between 58 and 67 °C in the Majimoto hot spring water. The thermal water convection mobilises elements. The amounts of $Ca^{2+}$ (0.26 mg l$^{-1}$) and $Mg^{2+}$ (0.24 mg l$^{-1}$) are comparably low, whereas $K^+$ (56 mg l$^{-1}$) is significantly higher than concentration measured in borehole water in the area (16 mg l$^{-1}$) (Maseke and Vegi, 2019).

Variations in lithology (Fig. 2a) are also expected to influence the hydrological conditions in the MRB. As such, because of their mostly low clay and organic matter contents, combined with high porosity and percentage of macrovoids, granitic soils have a poor water retention capacity. Volcanic soils however have a higher clay content and finer texture that they should be able to retain more water (Scalenghe, 2006). This is consistent with our findings in the spectral indices (Fig. 6), with lower vegetation coverage, lower clay mineral content and higher moisture stress in areas dominated by Precambrian granitoids. An

especially high moisture stress can be observed along the riverbanks of the middle reaches of the Mara River, when it crosses the Precambrian bedrock. As this section of the river runs through the Serengeti National Park, anthropogenic causes for the high MSI are unlikely.

Quaternary deposits in the area also show better conditions for plant growth. Due to a high amount of unconsolidated material and resulting high permeability, tectonically controlled alluvial fans could act as nutrient suppliers and might be areas of water

supply in the dry season.

The distribution of the various rock types has also influenced the large-scale drainage network, for example the resistant Quartzite ridges in the SE of the area dictate the flow of the rivers. Nevertheless, many streams flow across the strike of the basement rocks with only a slight lithological control. Stream patterns that were established on the lava cover were superimposed on the underlying basement rock (Williams, 1964a) when the lava was eroded by fluvial beveling.

**5.2.2 Clay mineral formation and distribution**

Clay is an important component of soil, as it binds nutrients and supports the water holding capacities of soils. Clay minerals can be formed over a wide range of environmental conditions and tectonic activity could influence clay forming and/or accumulating processes in a normal fault zone in various ways (Fig 9). The observed increased CMR values along mapped fault scarps (Fig. 6b) could indicate an accumulation of clay minerals in the faulted zone.

Physical clay mineral formation occurs in fault gauges, in the highly deformed zone of a fault by brittle deformation processes. During faulting, numerous fractures develop and open, which leads to enhanced weathering in the fracture zone. Authigenic



clays can be produced by chemical processes taking place during active faulting, such as feldspar dissolution through fluid-rock interaction (Solum et al., 2005; Yuan et al., 2019) or direct precipitation from circulating fluids in fractures in the hydrated rock mass and along the fault plane (Buatier et al., 2012).

In fault zones, clay minerals are not only formed in a fault gouge but also tend to accumulate along fault scarps through the enhanced exposure of weatherable minerals like feldspar or pyroxene and downslope transport across the escarpment (Birkeland, 1990). Weathering is enhanced on the footwall leading to soil degradation and leaching of minerals from the rock. The freshly exposed bedrock on the fault face is crossed by drainages transporting rocks and minerals downslope and is prone to landslides transporting material downslope (Eriksson, 1999; Ferrier and Perron, 2020).

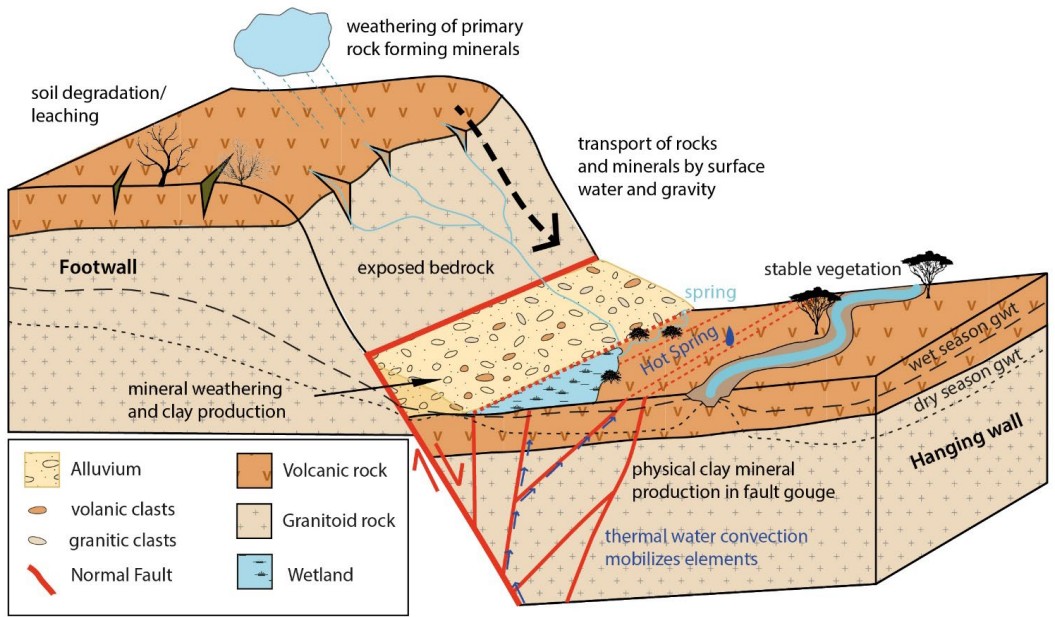


**Figure 7 Conceptual model of tectonic activity influencing hydrological and clay forming and/or accumulating processes in a normal fault zone and schematic distribution of different rock types in the study area.**

Clay-forming minerals in soils are an important factor in ecosystem stabilisation as they influence the retention of plant-available nutrients and water. Soils with a higher clay percentage are associated with a higher water-holding capacity, electrical

conductivity, and pH. They have higher cation exchange capacities (CECs) and 2:1 clay minerals provide plant available potassium, which often is a limiting macronutrient for plants (Barre et al., 2007). Also, clay–organic interaction stabilizes organic matter against rapid microbial decomposition (Yuan et al., 2021).

The observed lower CMR in areas dominated by granitic lithology compared to areas dominated by volcanics (Fig. 6b) can be explained by the mineralogy of the parent material. The Precambrian granitoid rocks in the MRB consist of quartz, sodic

plagioclase, potassium feldspar, biotite and hornblende, with lesser amount of chlorite, apatite and epidote. The Neogene lavas can be classified as trachytic phonolite composed of conspicuous glassy or white-weathered sanidine ($(K,Na)[(Si,Al)_4O_8]$) and greenish-grey nepheline ($(Na,K)AlSiO_4$) phenocrysts in a groundmass of sodic pyroxenes, amphiboles, nepheline and chalcedony (Grey et al., 1969). For the first phase of soil development, increasing age and weathering intensity lead to greater clay content and CEC values. Once the rock is heavily weathered, it becomes depleted in nutrients and soils become less fertile.

Jager (1982), found that soils developed on the Precambrian basement have a lower cation exchange capacity (140 - 290 mmol kg$^{-1}$) than those on the Neogene volcanics (ca. 440 mmol kg$^{-1}$). Through weathering, the rock-forming minerals decompose, and the micas, amphiboles, pyroxenes, and feldspars tend to weather to clay minerals, whereas quartz is resistant to weathering and builds the main constituent of the sand fraction of a soil. Granitic soils mostly contain kaolinite and





halloysite, largely derived from the weathering of feldspars, and vermiculitic minerals from micas (Wilson, 1976). Pyroxenes
and amphiboles in volcanic rocks are often altered to smectite minerals and even the sand fractions of the volcanic soils may
contain substantial quantities of smectite. With further weathering the smectite often becomes a mixture of kaolin and iron
oxide minerals (Scalenghe et al., 2006).

The concentrations of clay-forming minerals influence the retention of plant-available nutrients and water. We observed that
areas with a higher CMR mostly also show a higher NDVI and lower MSI value in the study area, indicating that the presence
of clay minerals enhances vegetation growth in the MRB. The distribution of clays can be linked to underlying lithology but
is also strongly controlled by faulting and by the resulting relative uplift and subsidence.

### 5.2.3 Impact on vegetation and soil fertility

When comparing the dry season and rainy season NDVI maps (Fig. 8 a, b) several locations with permanently high values
through different climatic conditions stand out, indicating healthy vegetation which are likely connected to tectonic processes.
Vegetation growth in the semiarid east African savannah is mainly regulated by soil fertility and rainfall which is linked to
environmental gradients like topography and geology (Bartzke et al., 2018; Desanker et al., 2020). There is a strong correlation
between stable high NDVI values and topography, which can be observed at the steep escarpments of both faults. A stark
contrast of high NDVI values along the hanging walls vs. low NDVI values along the footwalls of active faults is often
observed (Fig. 6a). Whereas high NDVI values at the active faults extend downslope for several kilometres over the steep
escarpments, high NDVI values at other – not tectonically controlled – cliffs are mostly limited to the cliffs themselves. This
indicates that tectonically active regions are more effective in producing larger stable vegetation zones than tectonically
quiescent regions by increased downslope transport of sediment and clay formation/accumulation.

The Mara Wetland is clearly visible on the dry season NDVI map (Fig. 8b) and in the time series (Fig. 8c), and shows constant
green vegetation with values from 0.6 to 0.75. The wetland downslope of the Utimbara Escarpment is a product of tectonic
subsidence and northward down-warping of the hanging wall, as well as uplift and northward tilting of the footwall.
Additionally, it is likely that secondary faults in the hanging wall combined with high rates of sedimentation and lake level
fluctuations of Lake Victoria lead to a high water retentivity and thus stable hydrological conditions. The interplay of these
factors created a climatically insensitive environment and possible drought refuge for the Serengeti migrants which today is
cut off from the migration route by anthropogenic alterations of the landscape (Peters et al., 2008).

Forested areas like the gallery forest along the Middle Mara River or the Trans Mara Conservation Area on the Isuria hanging
wall also show continuously stable NDVI values. The occurrence of dense gallery forest only along the Middle Mara River
could be due to the influence of tributaries from the forested Trans Mara Conservation area on the hanging wall, leading to
both, a lower MSI and a higher CMR for several kilometres along the river.

Agricultural areas on the Isuria and Utimbara footwalls as well as the Utimbara hanging wall show large variations between
dry and rainy season with values from 0.34 to 0.72 due to anthropogenic overprint. The expected tectonically induced soil
degradation on the footwall is not clearly visible in agricultural regions on the NDVI as plants on the croplands are sparser
than in the protected areas. The observed high NDVI values in the not tectonically controlled higher regions (Fig. 6a: Ntagare
Hills, Trans Mara Conservation Area, Mara gallery forest) though do not necessarily mean a higher current soil productivity
as soil chemical levels might still be lower as a result of soil degradation, while low lying regions receive more fresh material.
In order to accurately analyse the differences in soil fertility between footwall and hanging wall of both faults, a soil
composition analysis at various positions on footwall and hanging wall will be necessary in future field-based studies.

The largest NDVI variations between dry and rainy season can be observed in the Serengeti grasslands, which is likely the
result of a combination of strong rainfall gradients, overgrazing by large ungulate herds and wildfires (natural and
anthropogenically controlled) in the dry season.

Other natural factors also influence the NDVI pattern. NDVI has for instance been linked to rainfall by several authors (Reed et al., 2009; Richard and Poccard, 1998) concluding that there is a high correlation between vegetation performance and rainfall in Southern Africa. The high amount of rainfall in the Trans Mara Conservation on the footwall (1400 – 2000 mm yr$^{-1}$) in comparison to the Masai Mara Reserve on the hanging wall (800 – 1000 mm yr$^{-1}$) could compensate the lower soil fertility and lead to similar NDVI values despite different geological and morphological conditions.

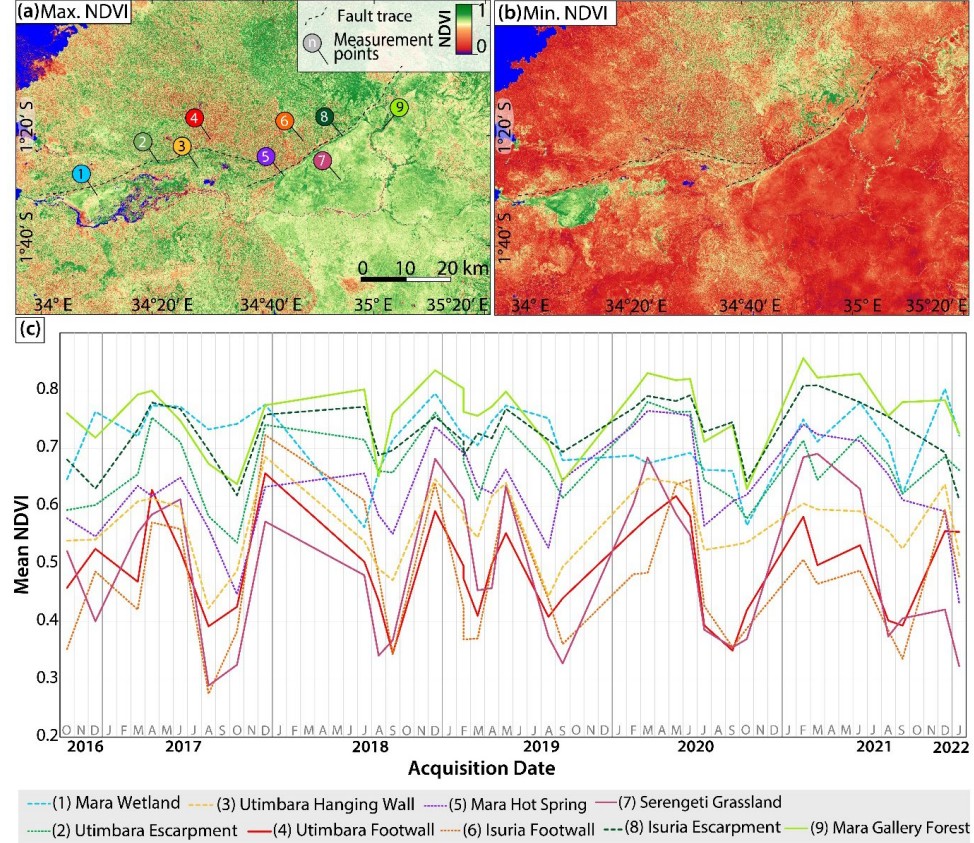


**Figure 8 Spatial and temporal distribution of NDVI along the IUFZ using a 5-year time series, derived from Sentinel 2 multispectral images from European Space Agency (ESA). Representative point measurements were conducted in nine selected areas for 34 Sentinel-2 scenes from October 2016 to January 2022. (a) Maximum NDVI derived from Sentinel-2 scene acquired on 28.12.2018 showing approximate positions of NDVI measurements (b) Minimum NDVI measured derived from Sentinel-2 scene acquired on**
**20.08.2017 (c) Mean NDVI time series for selected areas.**

Kübler et al. (2021) found that NDVI positively correlates with soil organic carbon (SOC) levels in the Southern Kenya rift. A similar correlation is to be expected in the MRB as the climatic and lithological framework is comparable. SOC is derived from plants and particularly, their roots with animals and soil microorganisms also contributing to the SOC stock (Lorenz and Lal, 2016).

Geological conditions, such as bedrock (parent material) lithology and erodibility strongly affect the quality and depth of soils and therefore, the distribution and seasonal stability of vegetation (e.g. Dahlgren et al., 2004). Several properties of bedrock could act as regulators of the distribution of vegetation. These include nutrient concentrations and concentrations of weatherable minerals which could limit plant growth. As weathering of primary rock forming minerals plays a prominent role in supply of base nutrients to soils, the chemical composition of bedrock and sediments strongly influences the total availability

of base cations (Ca, Mg, Na, and K) and heterogeneities in chemical composition result in variations in the soil productivity (Sinclair et al., 2008a). The geology in the MRB is highly variable, with nutrient-depleted metamorphic basement and Neogene





lavas with higher nutrient content (Olff and Hopcraft, 2008). Because of the significant age difference between the Precambrian basement and the Neogene volcanics, the basement rock is considerably stronger weathered and therefore contains less to no more weatherable minerals, indicated by the occurrence of laterite soils in the northern Serengeti (Mcnaughton, 1985). The

soils developed on trachytic phonolites show high Na-values (8-14%) but are lower than in the southern Serengeti and exchangeable K exceeded exchangeable Na. The granitic metamorphic rocks likely produce soils lower in Ca, Mg and plant-available P than the volcanic rocks. Another considerable factor are concentrations of possibly toxic elements in bedrock such as Co, Cu or Zn, which could inhibit growth (Arif et al., 2016).

The spectral measurements correlate well with the presumed lower soil fertility of granitic soils as areas covered by volcanic
rocks in general have higher NDVI values indicating healthier, dense vegetation (Kinyanjui, 2011). In order to display how NDVI varies along strike, the longitudinal profiles of both faults were extracted (Fig. 9). We selected three profiles along lines on top of the escarpment, in the sediment close to the foot of the escarpment and in the basin below to visualize vegetation differences between footwall, escarpment and hanging wall. We noted a marked difference in the NDVI profile shape between the Utimbara fault (Fig. 9a) as opposed to the Isuria fault (Fig. 9b). Further, we noted differences in the NDVI variance.

The western segment of the Utimbara escarpment is strongly degraded and dominated by an alternation of metamorphic bedrock and Quaternary sediment of the Mara Wetland, which explains strong variations in NDVI. The central segment of the Utimbara fault is dominated by a well-preserved escarpment with metamorphic basement on the footwall. Whereas the NDVI profile on the footwall and hanging wall show similarly low values, the NDVI along the foot of the escarpment is constantly high likely due to sedimentary input of freshly exposed bedrock from the steep escarpment. The eastern segment of the

Utimbara fault is a steep escarpment with Neogene volcanics on footwall and hanging wall showing variable NDVI values presumably due to anthropogenic overprint. At ca. 72 km distance along strike the escarpment foot profiles crosses the water-filled Mara gold, which explains the anomalously low NDVI value. The southwestern segment of the Isuria fault is dominated by a strongly degraded escarpment and higher NDVI values in the footwall than in the hanging wall likely due to volcanics on the footwall. The central segment of the Isuria fault is a continuously steep escarpment with volcanics on the footwall and

Quaternary sediments and alluvial fans on the hanging wall. From SW to NE the footwall and escarpment NDVI increases while the hanging wall NDVI decreases. This could be explained by the fact that the north-eastern segment of the Isuria fault is a degraded escarpment in Precambrian bedrock without nutrient-rich weatherable volcanic rock on the footwall.

Our observations imply that vegetation cover is to some degree controlled by underlying lithology and indicate that Neogene volcanics in general develop a denser vegetation cover than the Precambrian basement rocks.


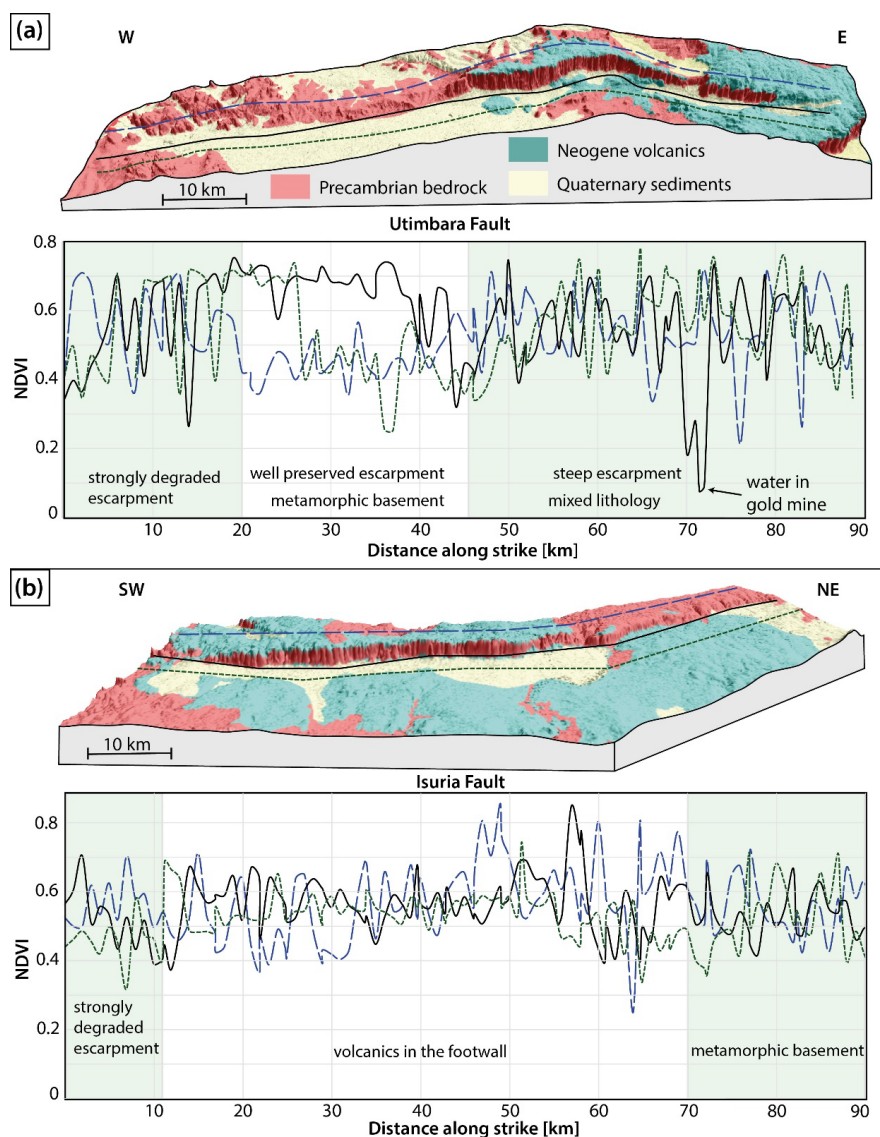

**Figure 9 Longitudinal mean NDVI (acquired between 2016 and 2022) profiles along (a) Utimbara and (b) Isuria fault. Profile traces are shown on 3D topography model derived from TanDEM-X digital elevation model © DLR 2019. Geological information derived from data by Geological Survey of Tanzania (GST). Note the changes of NDVI patterns related to geomorphological and lithological changes in the different segments of the escarpments as indicated by light-green and white colours in the NDVI profile.**

### 5.2.4 Animal subsistence

Sinclair et al. (2008b) analysed the dependency of ungulate migration patterns in the basin on rainfall and soil fertility. Their soil fertility map shows that the migration system is located on generally rich soils. They suggest that the western edge of the migration system approximates a transition to lower soil fertility originating from a geological transition from Neogene volcanics and Precambrian granites to nutrient depleted metamorphic rocks of the Mozambique Belt. To the north, the migration route of the Serengeti ungulates is inhibited by the topographic barrier created by the Isuria escarpment. Only during severe droughts on forage and water the herds moved to the grasslands up the escarpment (Sinclair et al., 2008b). This drought refuge today is no longer available for the ungulates as the landcover has been altered by humans in recent years.





Apart from determining the path of the large-scale animal migrations, geological conditions also influence the fauna in the
basin on smaller scales. Rocky outcrops like steep escarpments, tors, boulder-heaps, and insular domes (inselbergs) are widespread in the basin and create ecological niches for numerous specialized animals. The rocky outcrops support high levels of species diversity and endemism and can provide stable micro-climates for thousands of years (Fitzsimons and Michael, 2017). Woodland kopjes for example can conserve biodiversity in fragmented agricultural landscapes (Michael et al., 2008) and are inhabited by klipspringers (*Oreotragus oreotragus*) and rock hyraxes (*Procavia capensis johnstoni*) in the study area.
Moreover, they provide protected foraging sites for elephants and other large mammals (Sinclair et al., 2008b). Morrison et al. (2018) recorded a population growth of Serengeti elephants by up to 50.6% in the northern Serengeti along the escarpments. The tectonically controlled Mara Wetland is considered a critical refuge for a number of low oxygen tolerant fish species, such as the African lungfish (*Protopterus aethiopicus*) and provides a suitable breeding habitat for fish in Lake Victoria, such as the Smoothhead catfish (*Clarias liocephalus*) (Pringle et al., 2020).

**5.3 Anthropogenic vs. geological factors**

Our analysis of spectral vegetation indices in the area suggests that active faulting of the IUFZ improves conditions for vegetation along the escarpment. As most of the area today is influenced by human changes, separating the tectonic and geological signal from anthropogenic influences is a difficult task. Some of these patterns today are largely overprinted by anthropogenic activities. By systematic analysis of the spectral signals from a geological perspective though, natural factors
like lithological signals and increased erosion or deposition rates could be identified and delimited from anthropogenic signals. Strong differences between protected areas and agricultural regions are obvious on all three evaluated spectral indices, especially in the rainy season. This indicates a negative anthropogenic influence on vegetation growth, which tectonic and geological processes cannot buffer. Several studies examined for example the anthropogenic impact on soil erosion in the MRB and mainly attributed the increase to land use changes. The conversion of natural forest to farmlands and pasturelands
along the course of the Mara River provokes soil erosion and leads to a reduction in soil nutrients (Mati et al., 2008; Dwasi, 2002). Nonetheless, also tectonic activity could play a role in increased soil erosion.

**5.4 Environmental changes and their impact on stability of the Mara River Basin ecosystem**

The MRB is controlled by a delicate balance of tectonic subsidence related to the IUFZ and sediment input by the Mara River stabilizing the climatically sensitive ecosystem. According to climate change studies (Kendon et al., 2019; Shongwe et al.,
2011), the basin will experience an increase in annual river volume and rainfall amounts with wetter rainy seasons and drier dry seasons. This variability will result in higher peak flows in the wet period further increasing soil erosion and lower flows in the drier months leading to severe droughts (Osoro et al., 2018). Climate change in the area will likely lead to accelerated habitat desiccation and deterioration of vegetation quality for wildlife and livestock with the potential to disrupt the large-scale ungulate migration.
Today, there are several economic sectors that are strongly linked to a healthy MRB. The MRB supports tourism, agriculture and mining, which are three of the most profitable economic activities in Tanzania and Kenya, collectively contributing between 10-15% to both countries' Gross Domestic Products (Nelson, 2019). The Mara Wetland south of the Utimbara fault is the most valuable ecosystem service provider with a calculated total economic value of approximately 5 million USD a year with crop agriculture, water for commercial use, livestock and fisheries being the major contributors. The seasonally flooded
areas around the permanent wetland provide good grass for extensive livestock grazing and fertile soils for agriculture (Wwf-Esarpo, 2010). At least 73% of households around the Mara Wetland in Tanzania harvest fish for both subsistence and commercial purposes. Furthermore, the wetland provides important water cleansing services in the form of uptake of pollutants by species of Phragmites and Papyrus which could otherwise accumulate up the food chain. The plants can trap heavy metals in their roots and likely prevent heavy metal pollution reaching Lake Victoria (Matagi et al., 1998; Mati et al., 2008).



This balance is not only threatened by climate change, but also by anthropogenic impacts, such as an increase in sediment input, as analysed by Dutton et al. (2019). As the soil erosion on the footwall of the IUFZ is already increased due to tectonic uplift, an increase of the erosion rate on the footwall and deposition rate on the hanging wall could have serious consequences. Anthropogenic land use and cover changes progressively reduce the potential for wildlife to spread to future climatically suitable, tectonically stabilized habitats like the Mara Wetland (Ogutu et al., 2008).

**6. Conclusions**

Our results provide strong support for a major influence of geological processes on the MRB savannah ecosystem in the form of tectonic activity and lithological variations. Numerous neotectonic features, such as fault scarps in Mid-Pleistocene to Holocene sediment and down-tilt combing of the Mara River clearly demonstrate that the IUFZ has been tectonically active most likely during the Pleistocene and Holocene and shows compelling signs for recent surface rupturing activity. Future field-
based studies and dating of the offset volcanic rocks and deposition ages of alluvial sediments would enable determining precise displacement rates and better insight of the faulting history of this fault zone. Results on spectral time-series analysis presented here suggests that tectonic activity of the IUFZ perennially improves conditions for vegetation growth by trapping of water, rejuvenation of soils, clay mineral formation/accumulation, exposure of unweathered bedrock, and enhanced downslope processes along steep topography. Comparison of NDVI results with geological data demonstrates that the lithology
of bedrock also plays a considerable role influencing soil properties and thus vegetation cover all over the basin.
Our study shows that including geological factors and geomorphic knowledge in interdisciplinary ecosystem studies can significantly improve the overall understanding of other tectonically active regions in the world. Long-term insights on tectonically induced changes can be useful for recent management since the understanding of an area from a geomorphic perspective can complement natural processes within it.

**Author Contribution**

AL carried out geospatial and remote sensing analysis and wrote the manuscript with contributions and support from SK. SK supervised the project and carried out the study design. AL and SK interpreted the results.

**Competing interests**

The authors declare that they have no conflict of interest.

**Acknowledgements**

We thank Beth Kahle, Carolina Rosca and Mjahid Zebari for helpful discussions on the ideas presented in this paper. TanDEM-X data were kindly provided by the German Aerospace Center (DLR) through science proposal DEM_GEOL3221.

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
