# Peer review of "Tectonic controls on the ecosystem of the Mara River Basin, East Africa, from geomorphological and spectral indices analysis"

_Biogeosciences, 2022_

## Author Response (AR1)

Dear Prof. Dr. Anja Rammig,

We thank you for the time and effort spent in reviewing our manuscript. We are very pleased with the feedback received by the two reviewers and the editorial decision that allows us to submit a draft after minor revisions.

We thank Dr. Lydia Olaka and an anonymous reviewer for their positive feedback and helpful comments. We largely agree with the points raised and considered most of them in the revised version of the manuscript. Following Dr. Olaka's suggestions, we added additional information on the hydrological and clay-forming processes involved in geo-ecosystem interactions to the manuscript. Comments by the anonymous referee reinforced our decision to further confine the focus of the paper on the geo-ecosystem interactions of the studied fault zone and reduce the amount of detail with respect to the description of the fault structure and kinematics. We are confident that with the revisions made we have further improved the manuscript and by this, provide a valuable and novel contribution to the Biogeosciences community.

Yours sincerely,
Alina Ludat and Dr. Simon Kübler

In the following, our answers (blue) are listed below the points raised by the reviewers (black).

**Reviewer #1 (Lydia Olaka)**
**Reviewer #1**: The manuscript is largely well-written, but it needs improvement on the following sections:
The discussion section seems to omit to discuss the studies findings in context with other studies done in the EA Rift. e.g. Acosta, V. T., Schildgen, T. F., Clarke, B. A., Scherler, D., Bookhagen, B., Wittmann, H., ... & Strecker, M. R. (2015). Effect of vegetation cover on millennial-scale landscape denudation rates in East Africa. *Lithosphere*, *7*(4), 408-420.
Roller, S., Wittmann, H., Kastowski, M., and Hinderer, M., 2012, Erosion of the Rwenzori Mountains, East African Rift, from in situ–produced cosmogenic Be-10: Journal of Geophysical Research–Earth Surface, v. 117, F03003, doi:10.1029/2011JF002117.
**Authors**: We thank the reviewer for this valuable comment and the suggested literature. We have addressed this by adding further information on the relationship between vegetation, topography and erosion processes in the introductory paragraph (lines 45 – 48) and we have further added the suggested references.

**Reviewer #1**: Line 38 is not clear "sediments and water. Vertical surface motion relative to the water table leads to formation and/or modification of.."
**Authors**: To clarify the content, we have simplified the sentence to: "Tectonically induced vertical surface motion leads to modification of drainage systems and formation of hydrological features, such as inland lakes and wetlands."

**Reviewer #1**: Line 41," …channels that promote water circulation, resulting in hydrological surface features like springs and swamps in the vicinity of active faults". This is not correct, fault zones in the EAR are both conduits and can be barriers thus can also promote the disappearance of water sources when the faults are permeable e.g. Olaka, L. A., Kasemann, S. A., Sültenfuß, J., Wilke, F. D. H., Olago, D. O., Mulch, A., & Musolff, A. (2022). Tectonic control of groundwater recharge and flow in faulted volcanic aquifers. Water Resources Research, 58(7), e2022WR032016.
Authors: We thank the reviewer for pointing out this important aspect. We are aware of the function of faults as both hydrological barriers and conduits, as for example also shown in other rift settings, such as the Rheingraben in Central Europe. For our study area we hypothesize increased permeability along the active fault zones to be the dominant function influencing the local hydrological conditions. However, to point out the complex interactions between faults and groundwater, we have added additional information and references, as suggested (lines 38 – 43).

**Reviewer #1**: Line 110 Doming preceded rifting, this needs to be included here "…Rifting and subsequent rift shoulder"
**Authors:** We agree. A corrected version will include this change.

**Reviewer #1**: Line 279-280, the long term displacement timing of 0.10mm/y how was this determined because it is not mentioned in the method section? Additionally does this assume that the faulting has been gradual since 3.5-3 Ma. This is not clear. Rate of uplift
**Authors:** Information on the faulting history of the IUFZ is very sparse as we have explained in lines 280 - 291. With the available data we cannot provide more details on specific faulting episodes.

**Reviewer #1**: Line 275 -284 - seems like this is a discussion of the results within the results subsection
**Authors:** We feel like it is necessary to describe the tectonic setting in the results section, in order to be able to base the discussion of ecological processes linked to faulting on it.

**Reviewer #1**: Line 293 this section is not clear "The offset horizontal offset is mostly.."
**Authors:** A corrected version will include this change: "The horizontal offset is mostly.."

**Reviewer #1**: Figure 5. I would prefer to include the faults here too. Because the reader is supposed to associate the NDVI distribution especially for the dry season with the fault influence.
**Authors:** We have added the location of the IUFZ in the first panel.

**Reviewer #1**: Line 328 the reader is referred to the boundary of the Serengeti National Park in figure 6 which is not shown
**Authors:** The boundaries of the Serengeti National Park are shown in figure 2. We have also added the outline of the park to figure 5 and added a reference to the text.

**Reviewer #1**: Section 5.2.1: In the last 5 years there have been a number of studies on the fault kinematics of the East African Rift, and the tectonics of the Victoria plate. What is driving the deformation(subsidence and uplift (rate and pattern) on the two faults: Isuria and Utimbara? How are they influenced by regional tectonic activity.
See: Saria, E., E. Calais, D. S. Stamps, D. Delvaux, and C. J. H. Hartnady. "Present-day kinematics of the East African Rift." *Journal of Geophysical Research: Solid Earth* 119, no. 4 (2014): 3584-3600.
Stamps, D. S., Saria, E., & Kreemer, C. (2018). A geodetic strain rate model for the East African Rift System. *Scientific reports*, *8*(1), 732.
Ebinger, Cynthia J. "Recipe for rifting: Flavors of East Africa." (2021): 271-283.
**Authors:** We thank the reviewer for raising this important point. The internal deformation of the Victoria microplate is indeed a puzzling research problem. As previous studies have considered the Victoria microplate as a rigid plate rotating counterclockwise with respect to the Nubian plate (Glerum et al., 2020), providing additional information on active faulting in the interior of the plate is indeed of great importance for the active tectonics community. However, in this study we focus on the ecological impact of the IUFZ, rather than its wider tectonic implications, which will be addressed in a follow-up paper to be submitted to a more tectonically oriented journal representing another chapter of the first author's graduate thesis. (Ludat et al., in preparation).

**Reviewer #1**: Section 5.2.2: Could you comment on the difference between the clays formed around the thermal springs vs other locations. Does the existence of the thermal springs enhance clay formation in the area or not?
**Authors:** As described in lines 386 - 387 and visually shown in Figure 6b, the CMR shows increased values around the Majimoto Hot Spring. The nature and composition of the clay around the hot springs vs. other zones of clay accumulation cannot be determined with the chosen method. We have added this information to the manuscript (lines 387 - 388). With the aid of hyperspectral satellite data this question could be addressed and would be an interesting target for future research.

**Reviewer #1**: **Technical Corrections**
Line 74, Citation missing in the Reference list …Smith and Anderson, 2003).
Line 155, FAO should be in Caps
Line 292 instead of "drainages" consider replacing with drainage channel
**Authors:** Addressed and corrected all suggested changes.

**Reviewer #2 (Anonymous Referee #1)**

*Reviewer #2:* The manuscript uses TanDEM-X elevation data satellite imagery to identify and describe active normal faults at the margin of the Tanzanian Craton. Sentinel-2 multi-spectral imagery is then used to explore the relationship between faulting and the changing spatial and temporal distribution of vegetation. These relationships are further discussed in terms of the impact of tectonics on soil composition. The standard of presentation is very high with clear concise text and very high-quality illustration. I thought that the motivation and introduction were particularly well expressed. The observations presented are convincing both in terms of the fault activity and a clear link between the faulting and spatial patterns of vegetation. Overall, I found the manuscript interesting and thought-provoking and I believe that it could be published in its final version more or less 'as is'.

There are a few ways in which it could perhaps be expanded in terms of the description of the evidence of neotectonic activity and in terms of illustrating the role of anthropogenic activity/land use in controlling the temporal changes in vegetation. I point to a few of these suggestions below in my specific comments as well as indicating a few minor grammatical suggestions in my technical comments.

*Authors:* We thank the referee for reviewing our work and for the valuable comments that will improve the quality of our manuscript.

We also think that the neotectonic activity of the fault system deserves more attention and further detailed investigation. However, we have put the main focus of this study on the ecological impact of the IUFZ, rather than its wider tectonic implications to fit the focus of the Biogeosciences journal in the best way possible. To avoid further confusion about why we did not address this topic in more detail, we decided to only write about the IUFZ in general terms.

The detailed fault system and its kinematics will be addressed in a follow-up paper to be submitted to a more tectonically oriented journal representing another chapter of the first author's graduate thesis. (Ludat et al., in preparation)

We are happy to apply revisions to improve our manuscript as formulated in the answers to the referee comments below.

*Reviewer #2:* Lines 40-43: Or act as impermeable barriers leading to variations in water table depth…

*Authors:* We thank the reviewer for pointing out this important aspect. We are aware of faults acting as both hydrological barriers and conduits. For our study area we hypothesize increased permeability along the active fault zones to be the dominant function influencing the local hydrological conditions. However, to point out the complex interactions between faults and groundwater, we have added that they can act as impermeable barriers as suggested (line 41).

*Reviewer #2:* Line 122-123: It would be useful to include more detail on these 'smaller E-W trending faults'. Where are they? Do they affect the patterns of vegetation.

*Authors:* We thank the reviewer for raising this interesting point. The smaller E-W trending faults are exposed in Precambrian bedrock without showing signs of recent reactivation and also do not obviously affect the patterns of vegetation. This information has been added to the text (lines 127 - 128).

*Reviewer #2:* Section 2.1: I would consider discussing the presence or absence of instrumental seismicity within the region (and its mechanism – is there any evidence for an element of oblique slip?). It would also potentially be useful to briefly discuss the implications of recent geodetic studies: D.S. Stamps, C. Kreemer, R. Fernandes, T.A. Rajaonarison, G. Rambolamanana; Redefining East African Rift System kinematics. Geology 2021;; 49 (2): 150–155. doi: https://doi.org/10.1130/G47985.1

*Authors:* We thank the reviewer for this valuable comment and the suggested literature. We have added the suggested reference to chapter 4.1 and briefly discuss the implications of recent geodetic studies and instrumental seismicity (lines 280 – 281).

*Reviewer #2:* Lines 150-159: This distribution of land use appears critical to some of the later discussion. In the ideal world it would be possible to map the area under cultivation/under protection/forested etc. and to include this info in future figures. I realize this may not be possible though.

*Authors:* The distribution of landcover can be inferred from Sentinel-2 imagery shown in the background of figure 2. We have reduced the transparency of the layer to increase the visibility and the national park boundaries have been emphasized more clearly. Additionally, we have added further explanation to the figure caption: "Note the difference in land use inside and outside of the protected areas. While populated regions are dominated by agricultural land use, the protected areas are extensively covered by grassland and woodland type vegetation." (lines 134 – 136).

***Reviewer #2:*** Lines 289-296: Ideally I would like a little more detail here: Where are these scarps, how long are they, how do their heights vary though space, are they only in the small area highlighted in 3b,c? Do the apparent horizontal offsets have consistent sense of motion and magnitude? How large are they compared to the vertical offsets in the same section? The fact that these faults cut off from the main rift at a high angle suggests a significant oblique component seems quite possible and could be discussed further. In the final sentence what is meant by slope angle? Between what values do they vary?

***Authors:*** We acknowledge the demand for further information about the fault zone but as mentioned above, we have decided to exclude the details of the tectonics of the IUFZ, in order to put the focus on the ecosystem and provide the paper with a straightforward key theme that suits the scope of the journal best.

***Reviewer #2:*** Figure 5: More context (e.g. fault locations, park boundaries etc.) would help readers interpret these figures.

***Authors:*** We have added the location of the IUFZ to the first, and the national park boundaries to the first two panels.

***Reviewer #2:*** Sec 5.2.3 As mentioned above it would be useful to indicate more of the broad boundaries in land use on these figures – In particular the park boundaries and the edges of more intensely farmed regions. I suspect that in the more agricultural areas the results may be strongly affected by which areas of land are under cultivation – resulting in greater persistence of vegetation in areas where greater slope, and possibly other characteristics (e.g. very coarse alluvial fans) make them less attractive for agriculture. I am convinced that the fact that the relationship shows up very clearly in areas which are not under cultivation as well and so think that the conclusions drawn are valid, but it would be good to be able to assess differences between these areas more clearly.

***Authors:*** We thank the reviewer for this valuable comment and have addressed it, as mentioned above, by indicating the national park boundaries more clearly in figure 2. Additionally, the edges of more intensely farmed regions have been visualized using Sentinel-2 imagery and added to figure 2. To show seasonal differences in NDVI between cultivated and non-cultivated areas more clearly, national park boundaries were also added to figure 5.

***Reviewer #2:*** Figure 9: You might consider indicating the smaller scale variation of the geology along the profiles to better assess their impact on the variability. I find the differences between the profiles a little difficult to assess due to the variability.

***Authors:*** We thank the reviewer for this suggestion. We have further investigated the small-scale variations in lithology along the profiles and could not find a clear correlation between small-scale variations in NDVI and bedrock/parent material.

***Reviewer #2:*** **Technical Comments:**
Line 215-216: The comment on Savannah botany seems out of place here?

***Authors:*** We moved the sentence to chapter 5.2.3 Impact on vegetation and soil fertility, where it fits better into the context.

***Reviewer #2:*** Figure 2. Add reference for the boundary of Tanzanian Craton in inset.

***Authors:*** We have changed the figure caption to […] See small inset map of Eastern Africa for location and regional tectonic framework with boundary of the Tanzania Craton from Nyblade and Brazier (2002) […] (lines 131 – 132) and added the reference: Nyblade, A. A., and Brazier, R. A., 2002, Precambrian lithospheric controls on the development of the East African rift system: Geology, v. 30, no. 8, p. 755-758, https://doi.org/10.1130/0091-7613(2002)030<0755:PLCOTD>2.0.CO;2

***Reviewer #2:*** Line 293: remove one 'offset' and 'perpendicular' and state sense of motion.

***Authors:*** As we could not clearly determine the sense of motion of the fault, we have changed the sentence to: "The fault scarp segments are parallel to the escarpments and are frequently crossed by drainage channels, which show both seemingly sinistral and dextral horizontal offsets from 10 to 20 m when crossing the scarp (Fig. 3e)." (lines 300 – 302)

***Reviewer #2:*** Line 36: remove commas.
Line 69 & 157: MacClain/McClain?
Line 225: plant-available?

Shouldn't the first subsection in the discussion by 5.1 rather than 5.2?

Lines 590-591: 'the determination of'

Line 363-364: remove extra '.' and 'in general'

Line 499: McNaughton?

Line 554: However, through the systematic analysis of the spectral signals from a geological perspective, natural factors…

Line 591-592: 'Results of spectral time-series analysis suggest that the creation of steep topography through tectonic activity of the IUFZ perennially improves conditions for vegetation growth through the trapping of water, rejuvenation of soils, clay mineral formation/accumulation, exposure of unweathered bedrock, and enhancement of downslope processes.'

*Authors*: Addressed and corrected all suggested changes.

**List of all relevant changes made in the manuscript:**

- We have revised the colour schemes used in our figures to allow readers with colour vision deficiencies to correctly interpret our findings.
- We edited out some sentences describing our neotectonic observations as they are less important for our discussion of geo-ecosystem interactions (lines 306 – 309 of initial submission) .
- We added information on the hydrological setting of the Mara River Basin.
- Suggested references have been added to the manuscript.
- All detected typing and grammar mistakes have been corrected.